# Proof of Concept of Novel Visuo-Spatial-Motor Fall Prevention Training for Old People

**DOI:** 10.3390/geriatrics6030066

**Published:** 2021-06-29

**Authors:** Henk Koppelaar, Parastou Kordestani-Moghadam, Sareh Kouhkani, Farnoosh Irandoust, Gijs Segers, Lonneke de Haas, Thijmen Bantje, Martin van Warmerdam

**Affiliations:** 1Faculty of Electric and Electronic Engineering, Mathematics and Computer Science, Delft University of Technology, 2628 CD Delft, The Netherlands; 2Social Determinants of Health Research Center, Lorestan University of Medical Sciences, Korramabad, Iran; kparastou@yahoo.com; 3Department of Mathematics, Islamic University Shabestar Branch, Shabestar, Iran; skouhkani@yahoo.com; 4Department of Ophtalmology, Lorestan University of Medical Sciences, Korramabad, Iran; Farnoosh.Irandoost@yahoo.com; 5Gymi Sports & Visual Performance, 4907 BC Oosterhout, The Netherlands; gijs.segers@ziggo.nl; 6Monné Physical Care and Exercise, 4815 HD Breda, The Netherlands; Lonneke@Monne-ZorgenBeweging.nl (L.d.H.); valpreventiebreda@gmail.com (T.B.); 7Optometry van Warmerdam, 5211 KA ‘s-Hertogenbosch, The Netherlands; Martin@vanwarmerdam.nl

**Keywords:** balance disorder, older adults, falls, visuo-spatial-motor training

## Abstract

Falls in the geriatric population are one of the most important causes of disabilities in this age group. Its consequences impose a great deal of economic burden on health and insurance systems. This study was conducted by a multidisciplinary team with the aim of evaluating the effect of visuo-spatial-motor training for the prevention of falls in older adults. The subjects consisted of 31 volunteers aged 60 to 92 years who were studied in three groups: (1) A group under standard physical training, (2) a group under visuo-spatial-motor interventions, and (3) a control group (without any intervention). The results of the study showed that visual-spatial motor exercises significantly reduced the risk of falls of the subjects.

## 1. Introduction

Most fall accidents that occur to seniors happen at home, not during leisure but during domestic task behavior, i.e., by goal-directed behavior, with the exception of slips [1,2,3,4]. Home are filled with objects that are supposed to be in usual places, however if they are not then harm can be caused regarding non-perception of the trusted daily environment: Distraction/inattention by absence of mind, or stressed blinding of eyes by too many blinks and saccades.

The societal impact of this research is commonly understood from the cost of falling among older people (over 65) and subsequent hospital uptake. In the Netherlands, over 474 million Euros was spent in 2008 [5], which surged in 2018 to 960 million Euros [6], with over 6000 deaths. In other countries the picture is not different [7,8,9,10].

The societal impact to one’s psychological well-being is rarely embedded in frailty indices [11] and thus implicitly neglected, while it is in research on fear of falling [12,13,14,15,16,17] indicated to be of great importance

### 1.1. Falling of Older People: Dual Causes

A fall is unintentionally coming to the ground [18,19]. Delbare et al. [20] define ‘falling elderly’ as having had at least one injurious fall or at least two non-injurious falls during a 12-month follow-up period [21]. Most reported outcomes of falling [5,22] are: Concussion and broken bones (both 50%) and 20% loss of smell. A separate Physiopedia has been made for the problem [23]. The Journal of Safety’s special issue [24,25,26,27,28] and other journals on falls in older adults [21,29,30,31,32] reported many risk factors such as disability, medication, poor performance on physical tests, depressive symptoms, and memory of previous falls. The Cochrane Database reported reviews on interventions for preventing falls in older people living in the community in 2012 [33] and in hospitals in 2020 [34]. Memory may be a less obvious cause of falling, as it regards the fear of falling again [16,35]. Memories may change and can exaggerate fear [36].

Forgetfulness, i.e., the failing short term or immediate memory, is an independent risk factor for recurrent falls in persons aged 75 years [37,38]. Social life, posture, and balance disorders are related to health problems such as postural recovery [39,40].

The best test selection concerns balance-related impairments as critical predictors of falls [20]. Unfortunately, even old people with good balance may also become vulnerable to future fall risk because of disability by too low or exaggerated exercise level [20,41,42]. Merely asking one’s own risk of fall has predictive validity for the occurrence of repeated falls in older adults [43]. Yamada et al. [19] found the highest risk in 75–79 year interval, thereafter chances of falling decline and determine the usefulness of the trail walking test for predicting a fall. Lundin-Olsen [44] observed failing dual tasking of walking and talking as a predictor of falls, similar to Shumway-Cook in two papers [41,45]. Kim et al. [46] conclude that the SPPB and two dynamic balance test items of the Berg Balance Scale (BBS) can be used in screening for the risk of falls in an ambulatory older adult population. Singh et al. [42] tested physical performance against psychological factors [42] and found weak correlation results between PPA and physical performance tests such as TST, SPPB, FRT, TUG, and SBT. They conclude that physical performance may not be useful as a stand-alone test to screen for falls risk among community-dwelling older adults. This ties in with needed visual-spatial agility for older adults, as is the subject of this research.

The many correct ways to do the same movement [47,48] inhibits to single out one specific test movement as the best predictor for falls. Possible redundancy in learning new movements or improvements were studied by Furuki et al. [49] by using their decomposition method into relevant and irrelevant sub movements. These are the determinants of locomotor assessment in [50]. The difficulties of singling out best predictors is seconded by Balzer et al. [51] in a review of 184 publications selected from a database of 12,000 papers on fall prevention. They concluded that meta-analyses are not appropriate because of differences in research methods (for fall prevention in general). Previously in 2004, Chang et al. [52] had identified—from a number of health-related databases with thousands of papers—40 trials of interventions to prevent falls in older adults. They concluded that the most effective intervention was a multifactorial falls-risk assessment and management program. Pure exercise programs were less effective in reducing the risk of falling. The multifactor issue in this study caused by the neural system was foreseen by Woollacott in her editorial [53] on systems contributing to balance disorders.

Notwithstanding substantial differences between causes of falling as reported in the literature by scholars and self-report causes by 477 seniors, Zecevic [54] concluded that loss of balance was the leading cause of falls because in daily tasks such as cleaning the home, both the eyes and muscles have to perform simultaneously [55,56,57,58,59,60,61,62,63,64,65,66,67,68]. That is the visuo-spatial-motor system and motor system are operating concurrently [69,70]. However, the cerebellum uses both time and space separately, i.e., it has two systems for movements: A system for when to act and a system for where to act [71]. They are dual or even multiple, in the sense of cognitive loading [55]. Other Dual Tasks (DTs) studied in the research are verbal fluency [57], fine-motor movements [72], and arithmetic [73,74,75], with a review in [60]. Pijnappels et al. and Kannape et al. [76,77,78] pointed to gait changes during dual tasking, as a marker for age-related decline because these changes are more pronounced in older adults with fall risk. Currently, the gait is not used as such by physiatrists because of the rather lengthy series of measurements needed to get a precise diagnosis of instabilities, though EMGs can be helpful here [79].

Talking while walking is also dual tasking and therefore different [44,80] from solely walking [81]. Surprisingly, Kannape et al. [77] found that cognitive loading did not affect trajectory formation and its deviations, although it interfered with the participants’ walking velocity. This is because of the two different tasks of recognizing space versus navigating through it. This processing is performed by the same neural system [82]; the brain does not have dedicated and dissociable systems for each of these tasks. Only one system is to be trained, which is advantageous for old brains with some loss of connectivity between the eyes and brain. This confluence was conjectured by Jana et al. [83] and studied via simulation research [84,85].

### 1.2. A Medical Geriaters Wake-Up Call

Customarily do trainers exclude subjects with cognitive impairments because of the impact of even mild cognitive impairment or strokes on gait and balance [30,48,78,86,87,88,89]. Their argument is that movements requiring more information from the environment could be inhibited by sensory or cognitive impairment [90]. This common exclusion of impairments is under fire since the statement [91]—three years in succession—by gathered medical neurologists, geriaters, and other specialists that see frailty of older adults as caused by underlying symptoms and have stated not to treat aging as an independent process.

Earlier in 2007, Van der Velde et al. reported the effects of frailty in older adults if medications are stopped [92]. In 2013, Lee et al. [93] suggested interventions in cases of the TUG test giving abnormal results. In accordance with the medical geriaters call is the D-SCOPE (Detection, Support, and Care for Elderly) project [94] to identify factors that might influence the relation between frailty and positive outcome variables. An interesting proof of the geriaters’ viewpoint in the context of fall prevention can be inferred from Selinger et al. [95], whose team discovered that gait is optimized in real time. The inference is that a gait deviation has an underlying biological cause, which could be reverse engineered from the behavioral output. The geriaters viewpoint is also sustained by Arnadottir et al. [11] who found that sensory frailty is independent from motor ability associated with falls and problems in self-care. The sensorimotor system deteriorates with age and should be trained [96]. These results support our idea to complement the customary motor intervention by visuo-spatial-motor intervention.

Woollacott [53] foresaw that fall prevention not only is a motor activity but is also a cognitive activity, enabled by the plasticity of the brain. Saccades have evolved to help us protect from blurry images and keep our sight accurate, they have not yet adapted to the speed of our moving in the modern, motorized world [97,98]. To focus, our eyes typically shift in the direction of the object, which is a saccade. This causes a moment of inattentional blindness because the saccade masks sight [99]. While walking this poses few problems, but when driving down the roadway at 45 mph, the period of poor, peripheral sight, combined with saccadic masking can result in (even in the most conscientious driver) overlooking an object or a person. Magicians use this phenomenon to let even large objects ‘disappear’ [100]. This is a frequent problem with smaller objects, such as bicyclists who are hit by cars. Cyclists and other vehicles moving slowly in relation to the background are not salient in a driver’s peripheral vision and briefly disappear during the saccade. Metrics to detect older adults’ driver errors even for impaired cognition older adults are in [101], accompanied by an assessment of eye-tracking methods and technologies [102]. Elderly brains are even better equipped to discern movement at a further distance. Younger brains, however, are better at distinguishing movement nearby in the foreground because a younger brain is less sensible to motion in the larger background [103,104].

The medical 2018 wake-up call could lead to new training programs for old adults and patients with conditions such as schizophrenia, which has been linked to weaker motion segregation. Prior neural work paved the way to distinguish neural fields competing in visual perception versus dexterous command [105]. Such new training from motor to visuo-spatial-motor by DeLoss improved near acuity in older adults [106], sustained by later arguments from Nemoto et al. [107]. Pedroli et al. [108] combine cognitive and physical exercises in a VR biking test environment to successfully reduce the frailty of older adults. Ayed et al. [109] assessed via a case study the feasibility and effectiveness of prototype games on postural control and balance rehabilitation in a group of old people.

The first attempt to treat underlying symptoms of frailty by developing and evaluating whether mental combinatorial exercises confirm the geriatrics 2018 announcement was by Nemoto et al. [107]. Their study improved the visuospatial ability in older adults with and without frailty by pure cognitive training. Noohu [110] is exceptional in taking vision as a pillar of fall prevention. This is the basic tenet of this paper. The opposite issue has also been studied, with DeLoss [106] who aimed to prevent falling by improving vision via behavioral training. In this study, researchers do the opposite, with an introduction to visuo-spatial-motor training to prevent falls, supported by Feng’s research findings [70] that specialized neurons do violate our prejudice that movement comes after perception. These specialized neurons are activated by the intervention reported below as accessed by Diamond et al. [111].

### 1.3. A Visuo-Spatial-Motor Tool for Fall Prevention

Adults lose balance when their eyes are closed and space crew may lose their knowledge of limb and body position [112]. Eye reflexes (saccades) help the vestibular system to maintain balance [113,114,115,116,117] by rapid updates of the position and/or environment of the body to complete a push-off reaction [118]. Eyes have the fastest muscles of our body and express saccades are on top of these for rapid updates [113,114,115,117,119].

The neural system is so versatile that fear may cause an alteration of memories regarding falls, accidents, and movements [16,36,120,121], even if a subject has full balance, for instance a seated driver, eye movements with and without anxiety differ [122].

Selective impairment of balance if old people turn their head while walking [123,124] is another example of cognitive processing. The ability to distinguish inputs that are a consequence of our own actions (active motion) from changes in the external world (unexpected motion) is essential for perceptual stability and accurate motor control, but becomes worse in an older brain. At old age, balance is lost much more often than at a younger age because control of old vestibular systems might hamper [125,126]. Woollacott [53] explains experiments to induce a postural sway by a visual flow effect to test older adults’ balance stability. In general, does dual task complexity create a decision problem [127] for old brains and to what of the dual tasks does the brain give priority? For example, in the instance of if an older adult stops walking if they start talking. Diminished capability of dual tasking is one reason why Sherrington recommends walking for balance training, followed up by other researchers]. Recently, ref. [128] found that walking even enhances peripheral vision. Given that the lowest threshold is related to the vestibular system, deciding to choose the proprioception, somatosensory, and vestibular inputs is not a difficult task [129].

### 1.4. Blinks and Saccades Induce Perception Errors

It is known that old adults have greater eye movements than young people and there is, to a lesser degree, a corresponding pattern in brain activity, i.e., loss of cognition and interplay between gait, falls, and cognition is in [64,130]. On the other hand, saccades and micro saccades are well preserved in aging [131]. Visuo-spatial-motor training is to strengthen the connection of eyes with motor areas of the brain [75,132]. In addition, blinks contribute to the instability of a gaze during fixation because the eyes after a blink are not at the same spot [133,134,135,136,137,138,139,140]. Post-saccadic target blanking affects the detection of stimulus displacements across saccades in this way: Displacement detection is improved by blanks between views [133,137,138,141,142]. This contra-intuitive phenomenon is gratefully exploited in our visuo-spatial-motor training by carefully adjusting shutter frequencies. With each saccade do internal object representations change their retinal position and spatial resolution, which misleads peripheral views [132,143,144,145]. Perceptual continuity is a mental construct of the brain [146,147,148], even if eyes follow an object by smooth pursuit [149]. Perceptual illusion [150] occurs if the head moves, then heading is compressed [151,152,153]. The mature brain endows perspective upon space, with the role of foreshortening cues [154,155]. If, however, sight deteriorates with declines in contrast sensitivity and visual acuity [106,156,157] the continuity of perception and smooth pursuit decreases and the risk of falling increases, specifically if targets happen to move [158,159,160].

Fear of falling and memory of falls [161,162] interacts with perception by inducing restless saccades [122,163]. It is not the motor system that hampers because older adult eye muscles do this almost as well as younger adults, except for the stride which might become adapted for the fear of falling [13,164,165], recent studies of gait parameters [17,166,167] showed their importance. Restless saccades need optimization to reduce oversampling by viewers’ eyes, which hampers perception [168]. It should be noted that this is a potential achievement of our study.

### 1.5. Evaluation of Balance, Motor Skills, and Cordination

Both the exercise programs ‘Functional Walking’ and ‘In Balance’ were shown to be improving the scores on Tinetti’s Performance Oriented Mobility Assessment (POMA) [169] in the subgroup of pre-frail older adults. Faber et al. [170] tested the responsiveness of the POMA test for the prediction of falls with positive result. Besides the POMA, other tests have been advanced, such as the BBS [171], Functional Reach Test [172], Timed Up and Go [4,30,173,174], and a Clinical Test of Sensory Integration for Balance [175] to examine subjects’ ability to maintain quiet upright standing when sensory inputs change, and the Postural Sway measurements or Center of Pressure [176]. Podsiadlo and Richardson [177] introduced the timed version of the “Get-Up and Go” (TUG) in the original test by Mathias et al. [178].

To test capacity of predictability of the risk of falls in Northridge et al. [179] include vigorous subjects. Graafmans et al. [180], however concluded that mobility impairment is a predictor of falling. Shumway-Cook et al. [45] concluded TUG to be a sensitive and specific measure to discriminate fallers from non-fallers. This contrasts to the result in [42] that measuring postural sways (objectively using a balance board) is the only significant predictor of physiological falls risk among six tests. The cause argued in [176] is that the hypothesis of an intermittent velocity-based control of posture is more relevant than position-based control. To include velocities in tests goes back to Dzhafarov’s work [181,182,183,184] revealing that perception of velocity is a very different parameter from all other visuo-spatial-motor observations, which is the same with acceleration [97]. Kim et al. [46] concluded that the SPPB and two dynamic balance test items of the BBS can be used in screening for risk of falls in an ambulatory elderly population. Concluding from these pro and cons: TUG, POMA, SPPB, and HHD were included in the motor tests of this research.

## 2. Materials and Methods

### 2.1. Subjects

Participants were 31 volunteers, aged 60–92, from the client base of Monné Physical Care and Exercise and from the Pellikaan Fitness Center, both in the municipality of Breda, with a mean age of 77.85 ± 6.6 years and were assigned to three independent groups. Participants were free to decline any part of the protocol, except the tests. Most participants completed the interventions. Seven subjects left the program early, due to low motivation and personal problems, such as hospital uptake (2). The resulting group sizes were: No training (10), physical training (6), and visuo-spatial and motor training (11), during 12 weeks as in [107], extending the 8-week term of Paquette’s program [185].

### 2.2. Training and Test Instruments

For the visuo-spatial-motor intervention group and tests, we applied the wireless RGB LED powered lights that are included in the FitLight^®^ training system. These lights are used as targets for the user to deactivate as per the reaction training routine. Moreover, we used the Primary 2MJ^®^ stroboscopic spectacles (Figure 1, below) to train the sensorimotor system of subjects in the visuo-spatial-motor intervention group.

### 2.3. Data Collection

We applied a translation in Dutch of the MMSE recording information to the older adult subjects who participated voluntarily, and had not been diagnosed with dementia. We collected participants socio-demographic and health characteristics, as well as information about their past, including age, gender, marital status, presence of illnesses, disability status, fall history, fear of falling, drugs used, and walking habits. This form was created by the investigators and filled in by senior researcher LdeH together with every participant from all three groups of participants.

This study was performed in strict accordance with the recommendations of the Netherlands’ National Health and Medical Research Council statement on Ethical Conduct in Human Research. All procedures were approved by the Institutional Human Research Ethics Committees of the Lorestan University of Medical Sciences: Approval ID IR.LUMS.REC.1399.146, Korramabad, Iran. All participants gave written informed consent in accordance with the Declaration of Helsinki.

The tests before and after the chosen interventions were logged in Excel [186] and analyzed using Maple 2020 [187].

### 2.4. Research Development

Monné Physical Care and Exercise decided to introduce a renovated training program with embedded accredited training interventions for older adults in the Breda municipality, plus an experimental visuospatial module to specifically beat balance disorders for older adults. The idea of the visuospatial module is to evaluate the feasibility of such training as an add-on for accredited physiatrics treatment of balance disorders for older adults in the Netherlands (and abroad). We improved upon Nemoto et al. [107] by introducing a third group, a control group, on top of his locomotive and visuo-spatial-motor group. The control group is obviously not subjected to any intervention.

The visuo-spatial-motor-motor intervention in this research was performed upon invitation by the sport training expert (GS), because of his expertise in such training for athletes [105]. The research into effectivity of this renovated training program took place in 2019 and comprised of three groups: An observational control group, a group of trained by physical therapy (named the ‘motor group’), and a group trained by physical therapy + visuo-spatial-motor training (named the ‘visuo-motor group’).

The motor parts of the research program are based upon the Royal Netherlands Society of Physiatrics (KNGF) accredited many mobility programs [32] for older adults after a fall [188], from proven interventions [25,26,27,28,170,189,190] to effectively reduce the risk of falling, with exercises at least 3 h per week [191], even for the visually impaired [192]. Adapted names in the Netherlands are “In Balans”, “Vallen verleden tijd”, “Zicht op evenwicht”, “Bewegen valt goed”, and “Otago training”. The group training “In Balans” includes an explanation of causes of falling and reflection upon own movements, inspired by Tai Chi. We decided to take a mix from all of these for motor intervention.

#### 2.4.1. The Eyes as a Tool for Maintaining Balance

In this research were applied the Japanese Shutterglass Primary 2MJ^®^ (Figure 1). A support for our visuo-spatial-motor approach was from Coubard et al. [193]: Fall prevention modulates decisional saccadic behavior in aging. For instance, the 60-s test in Figure 2 collects the reaction times of quick hand movements aimed at extinguishing FitLights^®^ mounted at a window.

For multisensory experiments, video toolboxes were designed e.g., to transform numbers into movies [194,195]. This fulfills in a simpler way our need, than virtual reality (VR) equipment would for our new training, see for specification Figure 3 and Appendix B, below. Molina et al. [196] and Mirelman et al. [197] applied the idea in an immersed virtual reality, also introduced as Exergaming [198,199]. Our rationale was that traffic signs are of a different nature than other images in everyday life. This perception was trained by our team specifically by way of a set-up with FitLight^®^s on a table, if one of them lights up then it has, as fast as possible, to be touched by a hand. Then the visual focus had to change swiftly to the lights and numbers in the distance, as depicted in the photographs of Figure 3 (or Figure A2 of Appendix B). The distant lightning color had to be named and directly thereafter the number beneath it. The swift change of view distance entrained fusion flexibility of sight.

#### 2.4.2. Peripheral Sight

To train peripheral sight FitLight^®^s were used on the front view of a window (Figure 2), on a table (Figure 3), and on the floor as photographed in Figure 4. The peripheral sight is challenged because of the demand to look forward and to perceive the lights in the periphery (Figure 2), on the table (Figure 3), or floor (Figure 4, below).

### 2.5. The Interventions

A 12-week program from January 2019 to April 2019 included a mix of the trainings given in the accredited Netherlands programs, plus our new visuo-spatial-motor program. The arguments for the set-up are in this section and in more detail in Appendix A, Appendix B and Appendix C. The participation and small sample size are in Section 2.1.

#### 2.5.1. Physical Exercises, the Motor Program

Motor-based tests as discussed in Section 2.2 for the ability to prevent a fall by keeping balance and control [1,47,48,110,200,201,202] or by improving postures and attitudes [2,203] are embedded in traditional motor training programs. Balance-impaired older subjects were assessed by Cho et al. [4]. Associations between performance in the TUG and the Six-Minute Walk Distance (6MWD) with physiological characteristics were researched by Montgomery et al. [204]. The result was an appendicular lean muscle mass percentage indicator for women in the TUG performance and for men, their jump power. The subjects in this research were in the mean 77.8 years. The set-up of the motor part of this research is a mixture of exercises embeddable for the tests in Section 2.2.

#### 2.5.2. Visual Plus Physical Exercises: Visuo-Spatial-Motor Intervention

If supplemented by stroboscopic spectacles, trainees alter their perception of movement [165]. This enabled the starting point for the newly-developed intervention. Heindorf et al. [205] demonstrate that the motor cortex mediates corrective behavioral responses to unexpected visual perturbations by not ‘simply’ controlling movement, but the sensory guide control of movement in instances where the sensory processing was unknown and therefore dependent of cortical processing. Though keeping balance is automatic and/or anticipatory, aging and vision loss both decrease fitness to tell if we are moving or if we see a moving object.

In our study the researchers [206] designed an entropy index to distinguish eye movements between erratic saccades or normally wandering eyes. The entropy index enables discrimination between erratic and common saccades. It is understood that from sports training, expert players have lesser eye movements than unexperienced players [207,208]. The same holds for older adults, therefore was the intervention interspersed with short exercises of throwing balls while wearing the Primer 2MJ stroboscopic spectacles. A few minutes suffices to enhance the sensorimotor stamina of older adults. For tests of the achieved performance, we applied FitLight^®^ signaling in two ways: With static time delay and with dynamic (changing) time delay.

### 2.6. Statistical Analysis

To exclude bias among the grouping of trainees, we analyzed the grouping in Section 2.6.1. by a Chi-Square independence test for all groups to find that for the statistically significant independence sampling of all three groups the significance level was 0.05 (*p* < 0.05). In Section 2.6.2 and Section 2.6.3, the effects of the experiments are reported via the differences between the pre-tests $T_{0}$ and the post-tests $T_{1}$.

#### 2.6.1. Testing Independence of the Three Intervention Groups

The three different groups named, motor, visuo-spatial-motor, and control, of older adults with about the same ability and age were tested against the null hypothesis that the three groups are the same, i.e., sampled from the same population, i.e., statistically characterized by one multinomial distribution. The independence is needed for the three groups w.r.t. the administered interventions: Motor, visuo-spatial-motor, and control.

For independent testing of the groups assigned to the interventions, we required both the mean pre-test scores in Table 1 and the mean post-test scores in Table 2.

The outcomes $T_{0}$ in Table 1 of the prior SPPB, TUG, and POMA tests and the subsequent posteriors $T_{1}$ in Table 2 of these interventions are statistically tested by the Chi^2^ independence test [209] at a 5% significance level, up to 5 decimals of accuracy, for readability maximally two decimals are displayed in Table 1 and Table 2. Three intervention groups had SPPB, TUG, and POMA for pre- and post-tests both. This makes together (3–1) times (6–1) = 10 degrees of statistical freedom. The computed statistic is 2.71210, far below its critical value 18.3070, with a probability of p=0.987411. Which does not provide enough evidence to conclude that the null hypothesis is false. The independence test of the three groups of our subjects w.r.t. to visuo-spatial-motor tests is similarly done by the Chi^2^ independence test [209] at a 5% significance level, up to 5 decimals of accuracy. The summary of outcomes is displayed in Table 3.

#### 2.6.2. Comparison of Intervention Groups w.r.t. the SPPB, TUG, and POMA Motor Tests

The motor intervention as tested by SPPB, TUG, and POMA requires the number of hits or seconds of time elapse. The TUG is the timing test among the three motor tests. This explains why the fitted regression lines are in Figure 5 below the neutral line: If the intervention has a positive effect then the resulting regression line has a direction coefficient lower than 1, i.e., below the black neutral line in Figure 5.

#### 2.6.3. Comparison of the Interventions w.r.t. the Visuo-Spatial-Motor Tests

Instead of absolute measurements (hit counts and/or timing values) such as the above for the SPPB, TUG, and POMA in Figure 4, Figure 5 and Figure 6, we display the differences between successive scores of the visuo-spatial-motor performances of subjects. This gives an immediate picture of progress, or deterioration, as shown in Figure 7, Figure 8, Figure 9 and Figure 10. To use differences instead of the raw measurements is a method borrowed from physics to display states of ensembles of particles as it gives an immediate overview of what has happened.

To review this idea at the hand of the tests with static FitLight^®^s: Figure 4 shows at the horizontal axis the elapsed time in the post-test minus the elapsed time in the pre-test, i.e., $T_{1}-T_{0}$. The vertical axis displays the count of the number of hits at time $T_{1}$ minus the count of the number of hits at time $T_{0}$. So, vertically it has the improvement (or decline) of the number of hits within the time gain depicted at the horizontal axis.

We scored improvement of gain with positive numbers. So, in Figure 8, Figure 9, Figure 10 and Figure 11 are the improvements of reaction times is depicted to reach a number of hits. Then at the horizontal axis, there is the gain (i.e., the reduction) in the Reaction Time (RT), against the vertical display of the gained number of hits within time.

A neglected parameter in tests of fall risk is the visuo-spatial-motor component of acting to prevent falling. We tested the research hypothesis that a visuo-spatial extension of training to prevent falling did not have any effect.

The data of these small groups were fitted with regression lines according the trimmed least squares method. This method optimizes the residual error in the fitting procedure to the least possible given value for the dataset at hand.

## 3. Results

A concise and precise description of the experimental results, their interpretation as well as the experimental conclusions will be drawn in this chapter. We did not group or adjust for age and sex or corresponding baseline values of data.

### 3.1. Independence of the Groups

In our analysis in Section 2.6.1 we tested the hypothesis $H_{0}$ that the three groups showed no difference effect with regard to the motor testing methods SPPB, TUG, HHD, and/or POMA (see Table 1 and Table 2). The researchers of this study left out the HHD measurements because they were invariant over prior- and posterior testing, with the exception of only one subject with a small deviation between pre- and post-test.

We tested the hypothesis $H_{0}$ that the three groups showed no difference effect with regard to the visuo-spatial-motor testing methods with FitLight^®^s, peripheral view, and fusion flexibility. The outcomes are listed in Table 3.

Overall, the result for all the tests, without exception, is that the null hypothesis is not rejected under 5% level of significance, and the groups are not dissimilar.

In conclusion, the hypothesis showed that the grouping of clients is effective and cannot be refuted on basis of these pre- and post-intervention tests.

### 3.2. Summary of Section

#### Comparison of the Intervention w.r.t. SPPB, TUG, and POMA

In the SPPB test of Figure 5 is the motor group, i.e., the green regression coefficient, lower than the blue line of the visuo-spatial-motor group. This says that the motor group is slower than the other groups up to a pre-test score 8. From this we conclude that the break-even point of motor versus visuo-spatial-motor intervention lies at the SPPB initial score of about 8. For the POMA test in Figure 7, it seems to reign the opposite with a break-even point at a score of 24.

### 3.3. Comparison of Interventions and Subjects with Help of Testing with FitLight^®^s

The visuo-spatial-motor interventions were also tested. Output is in Figure 8, Figure 9, Figure 10 and Figure 11.

The curves of the three subject groups in Figure 8, Figure 9, Figure 10 and Figure 11 depict the gained speed on the horizontal axis and the gained nr. of hits on the vertical axis. The best performances are by visuo-spatial-motor training. A maximum score is achieved by a female subject of 92 years (at the top of the diagram in Figure 8).

The peripheral step test has about the same slopes for groups with the slopes in Figure 9 and Figure 10 close. We measure the angle beteen slopes $s_{1},\;s_{2}$ pairwise by the mathematical cosine measure $\cos(s_{1},\;s_{2})$. If the cosine is 1, the interventions are similar w.r.t. the administered test. If the cosine is 0, the interventions are dissimilar w.r.t. the administered test. This degree of similarity is a diagnostic tool for the interventions as a whole, not for the individuals subjected to it. The subjects might individually score very poor on a test, such as in Figure 11, the motor group only had deteriorated visual stamina after the intervention (see the circles all in the third quadrant of Figure 11), while the visuo-spatial-motor group only had an improved visual stamina after the intervention (see the rhombus points all in the first quadrant of Figure 11).

This is remarkable and clear because of the very discriminatory trait of the two interventions for the subjects, as the interventions itself indifferent with respect to the Peripheral Vision test (Figure 10) and the Fusion Flexibility test (Figure 11).

Concluding, both the Peripheral Vision and the Fusion Flexibility test are equally applicable for motor intervention and visuo-spatial-motor intervention. Moreover, the two tests are very discriminatory for subject performances.

Both the fusion flexibility and peripheral test splits the performance of subjects in the motor group and the visuo-spatial-motor group in two very different regimes, as can be seen in Figure 10 and Figure 11: Motor intervention and control subjects score on the negative horizontal and vertical axes, i.e., the third quadrant of the coordinate plane. This means that the gained number of hits in the post-intervention test is lower than in the pre-intervention test. This result is quite the opposite of the result of the subjects of our novel visuo-spatial-motor training. These subjects only score in the first quadrant at the top right of Figure 10 and Figure 11, i.e., along the positive coordinate axes. The logic of this opposed effect is evident, as seen from the literature.

## 4. Discussion

If, in the absence of stress, the perception of objects is uncertain then ‘rehearsing’ by repeated saccades [210,211,212,213] is to reduce uncertainty in perception. To remember a phone number, we may rehearse the digits mentally. Eyes do automatically something similar to help recall what we see in sequence when we are old [214,215]. When remembering becomes difficult, eye movements also help to see the world as an external memory [216]. This postulated embodied cognition [217] assumes that instead of storing visual information in working memory, information is retrieved by appropriate eye movements [100,214,218]. These reflexive saccades for sensory attenuation [219] increase with age. As a result, older adults have a greater reliance on predictive than on sensory signals [220,221,222]. It becomes predictive because of fear [16,35,64,93,121,161,162,223,224]. This is a reason to do visuo-spatial-motor training as it improves perception, which becomes at rest after such training. The study of eye movements helps one to know if such movements become erratic and the brain loses ‘sight’ [202,225]. This is known from stressful situations, such as in athletic field games (hockey, baseball, and football) and from disturbances during space flight [112,226]. To this end we initiated an entropy tool for the visuo-spatial toolbox [206].

Visuo-spatial-motor training is nowadays ubiquitous in enhancing athletes’ abilities [105]. The asset of visuo-spatial-motor training is: shutting down the view has the effect of saccades performing slightly poorer, which increases the saccade size [227]. Wilkins and Appelbaum [228] review the variants of the trainings as performed over the globe, however, their application of Senaptec spectacles is not fully embeddable for training of older adults because of its small range in shutter frequencies.

The visuo-spatial-motor training evokes hidden and/or underdeveloped signal queuing by forcing the older adults brain out of its comfort zone, as reviewed by Liu et al. [130], though they recommend pure cognitive regimens (e.g., video game training) to reduce the incidence of falls. The research and results presented here follow a motor-based avenue and add to the results by and confirm Nemoto et al.’s [107] conclusion that motor plus visuo-spatial exercise is a feasible exercise program to potentially improve visuo-spatial ability and overall cognition in older adults with and without frailty. Nemoto et al. compared their visuo-spatial intervention to 13 other programs. Only gait speed did not improve by their visuo-spatial intervention. This is seconded by Pijnappels, Rispens, and van Schooten et al. [229,230,231] because the capacity to generate maximum extension force by the whole leg (e.g., in a leg press apparatus or during jumping) results in the best discrimination rule between older fallers and non-fallers. This capacity is out of reach for visuo-spatial-motor training as reflected in our result in Figure 4 where vigorous subjects above SPPB is 8 and do not profit as much as from the visuo-spatial-motor intervention compared to subjects of motor intervention.

## 5. Conclusions

The results in Section 3.1 all indicate that the groups as stratified between physiological training, visuo-spatial-motor training, and a control group do not differ w.r.t. the administered tests.

Our approach to augment the reality of trainees during brief time intervals from 3 to 15 min by visuo-spatial-motor intervention, and to not replace reality, worked well to raise the agility of subjects’ mind and eyes. This confirms earlier findings with athlete subjects [105]: The phenomenon of a quiet view as if traffic is moving slower than seen before the training. In general, this lasted about 24 h. We enhanced their reaction time and visual stamina (Section 3.3) and subsequently reduced the risk of falling to a large extent by enhancing the sensorimotor system of older adults by our new type of visuo-spatial-motor add-on to complement traditional omniscient physical training. From the initial inspection, subjects had a hampered stride, however after 12 weeks of the visuo-spatial-motor intervention they walked freely and independently.

## Figures and Tables

**Figure 1 geriatrics-06-00066-f001:**
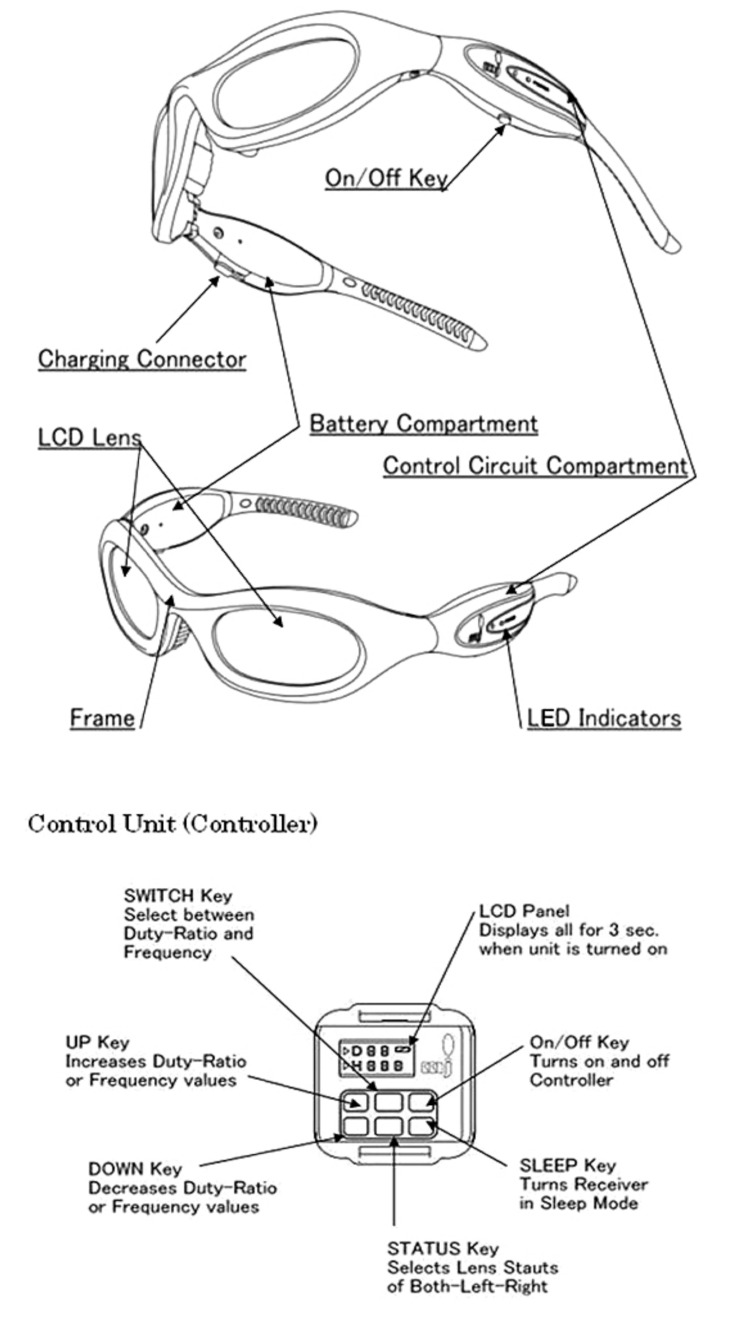
Shutterglass Primary 2MJ^®^ with a wireless control unit to enable visual resetting.

**Figure 2 geriatrics-06-00066-f002:**
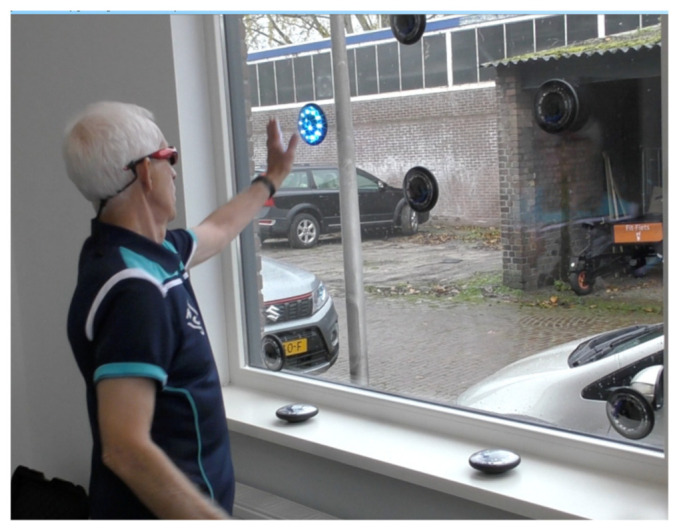
FitLight^®^s mounted on a window for testing reaction time. The specification of the experimental set-up is in Appendix A.

**Figure 3 geriatrics-06-00066-f003:**
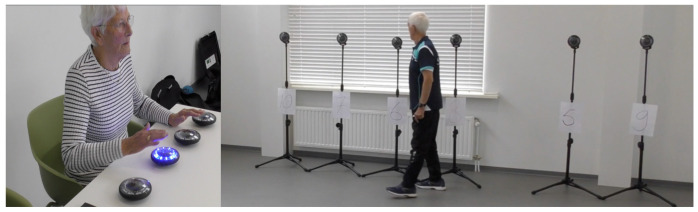
FitLight^®^s for testing eyes’ fusion flexibility by the reaction time of saccades from the table redirected to the distant lights and numbers. The specification of this set-up is in Appendix B below.

**Figure 4 geriatrics-06-00066-f004:**
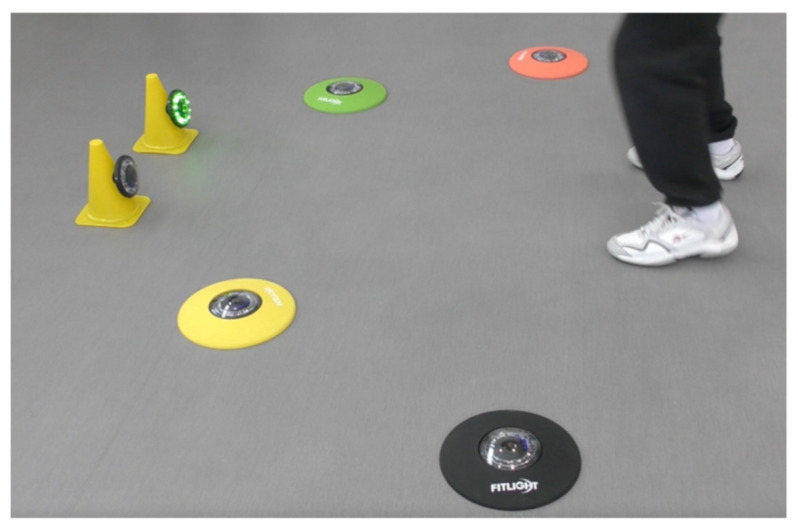
FitLight^®^s embedded in colored rings. Subjects stand in between to dampen a light by moving a leg over it or close to it. The experimental set-up is displayed in Appendix C.

**Figure 5 geriatrics-06-00066-f005:**
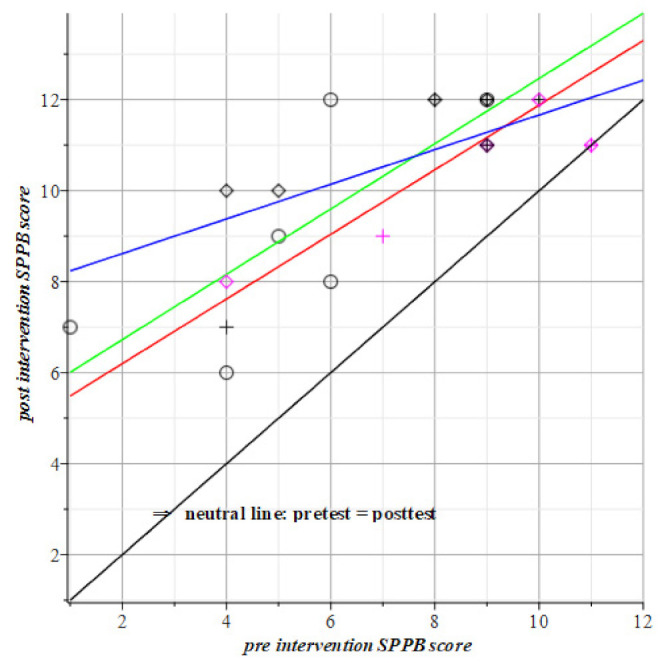
The mean results of the motor intervention are green; the visuo-spatial-motor intervention mean line is blue; the red line is the control group; the no-results, or neutral line is black. All groups are above the black line, so they all made progress, compared to their respective pre-intervention scores. The visuo-spatial-motor group performs better until break even (SPPB = 8). The motor group performs better after SPPB = 8. Red datapoints represent male subjects. Circles and green line: Motor group, $T_{1}=$ 5.29 (±1.81) + 0.72 (±0.31) $T_{0}(SPPB)$. Rhombus pts., blue: Visuo-spatial-motor group, $T_{1}=$ 7.86 (±0.95) + 0.38 (±0.12) $T_{0}(SPPB)$. Plus signs, red line: Control group, $T_{1}=$ 4.78 (±1.56) + 0.71 (±0.18) $T_{0}(SPPB)$.

**Figure 6 geriatrics-06-00066-f006:**
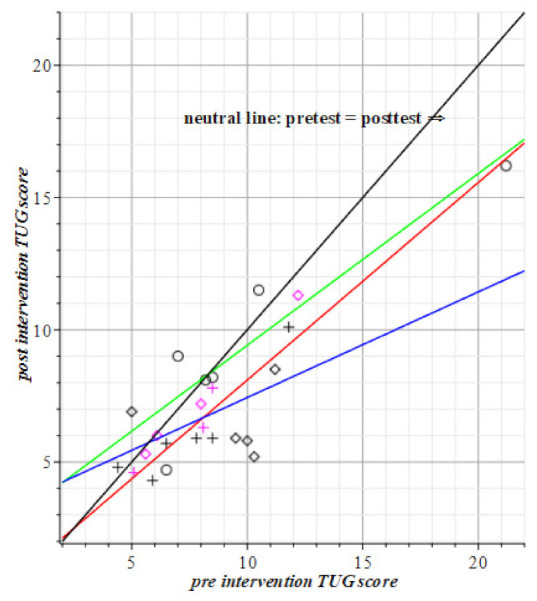
The mean results of the motor intervention are green; the visuo-spatial-motor intervention mean line is blue; the red line is the control group; the no-results, or neutral line is black. All groups are below the black line, so they all made progress, compared to their respective pre-intervention scores. The visuo-spatial-motor group performs the fastest (in the mean). Red datapoints represent male subjects. Circles and green line: Motor group, $T_{1}=$ 2.92 (±1.60) + 0.65 (±0.14) $T_{0}(TUG)$. Rhombus pts., blue: Visuo-spatial-motor group, $T_{1}=$ 3.44 (±2.17) + 0.40 (±0.24) $T_{0}(TUG)$. Plus signs, red line: Control group, $T_{1}=$ 0.63 (±0.94) + 0.75 (±0.12) $T_{0}(TUG)$.

**Figure 7 geriatrics-06-00066-f007:**
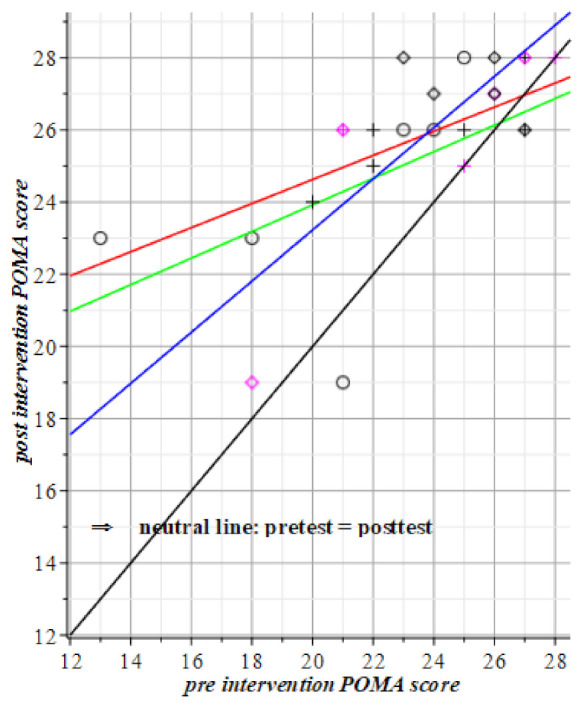
The results of the three groups in green, blue, and red lines. The no-results, or neutral line is black. All groups are above the black line, so they all made progress, compared to their respective pre-intervention scores. The visuo-spatial-motor group performance shows a break-even point at about $T_{0}$ = 22. This means that initially, vigorous subjects perform better at POMA after the visuo-spatial-motor intervention than vigorous subjects would profit from the only motor intervention. For frail subjects w.r.t. the POMA score result is opposite: Until intake score $T_{0}$ = 22, the motor intervention only is more beneficial than the visuo-spatial-motor intervention. Red datapoints represent male subjects. Circles and green line: Motor group $T_{1}=$ 16.55 (±1.82) + 0.37 (±0.31) $T_{0}(POMA)$. Rhombus pts., blue: Visuo-spatial-motor group, $T_{1}=$ 9.05 (±5.36) + 0.71 (±0.22) $T_{0}(POMA)$. Plus signs, red line: Control group, $T_{1}=$ 17.95 (±0.77) + 0.33 (±0.33) $T_{0}(POMA)$.

**Figure 8 geriatrics-06-00066-f008:**
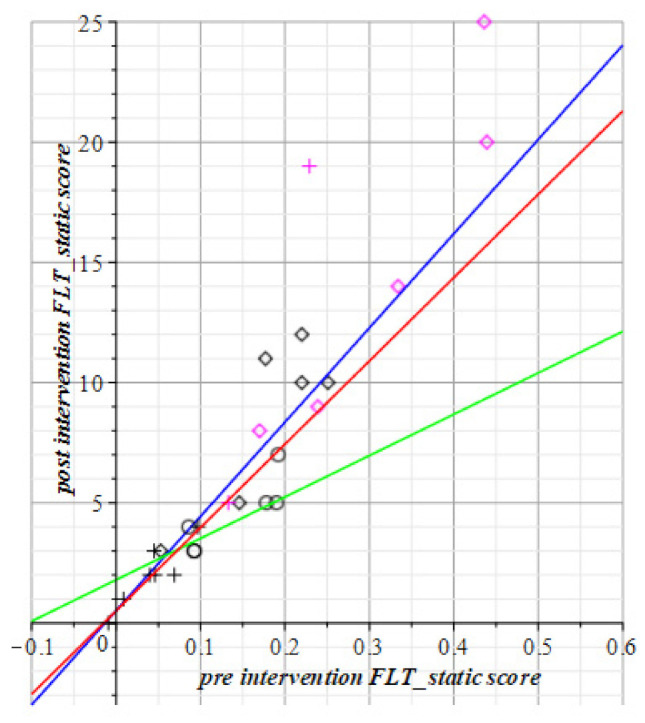
This depicts the results of the static FitLight^®^s test in the number of hits per unit time interval. The + points are the control group (Red line); diamond points are the vision group (blue line); and circles are the motor group (green line). Both the visuo-spatial-motor and control group have an outlier. Red datapoints represent male subjects. Circles and green line: Motor group $T_{1}=$ 1.80 (±1.09) +17.2 (±7.4) $T_{0}$. Rhombus pts., blue: Visuo-spatial-motor group, $T_{1}=$ 0.52 (±1.59) + 39.2 (±5.9) $T_{0}$. Plus signs, red line: Control group, $T_{1}=$ 0.51 (±1.14) + 34.7 (±11.4) $T_{0}$.

**Figure 9 geriatrics-06-00066-f009:**
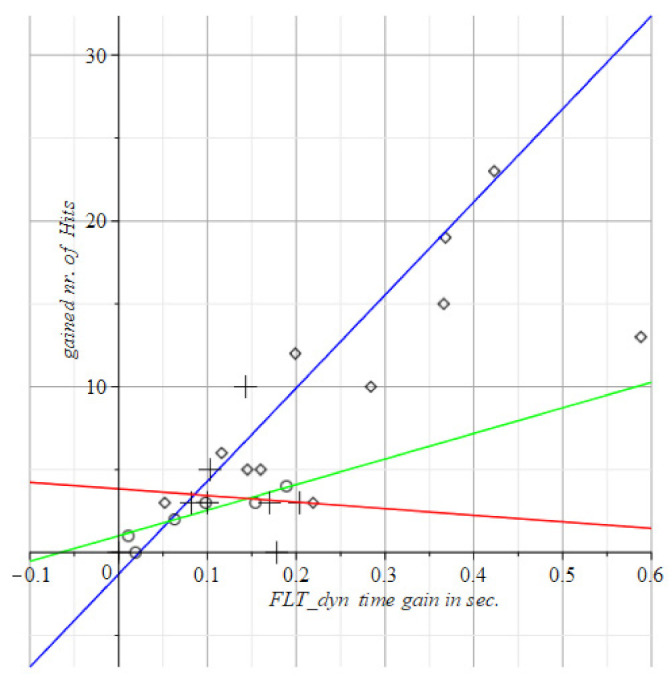
Depicts the results of the dynamic FitLight^®^s test. This test requires rapid eye movements, hence the results of the blue line are best. Circles and green line: Motor group, $T_{1}=$ 1.80 (±1.60) + 17.2 (±0.14) $T_{0}$. Rhombus pts., blue: Visuo-spatial-motor group, $T_{1}=$ 0.52 (±2.17) + 39.2 (±0.24) $T_{0}$. Plus signs, red line: Control group, $T_{1}=$ 0.51 (±0.94) + 34.7 (±0.12) $T_{0}$.

**Figure 10 geriatrics-06-00066-f010:**
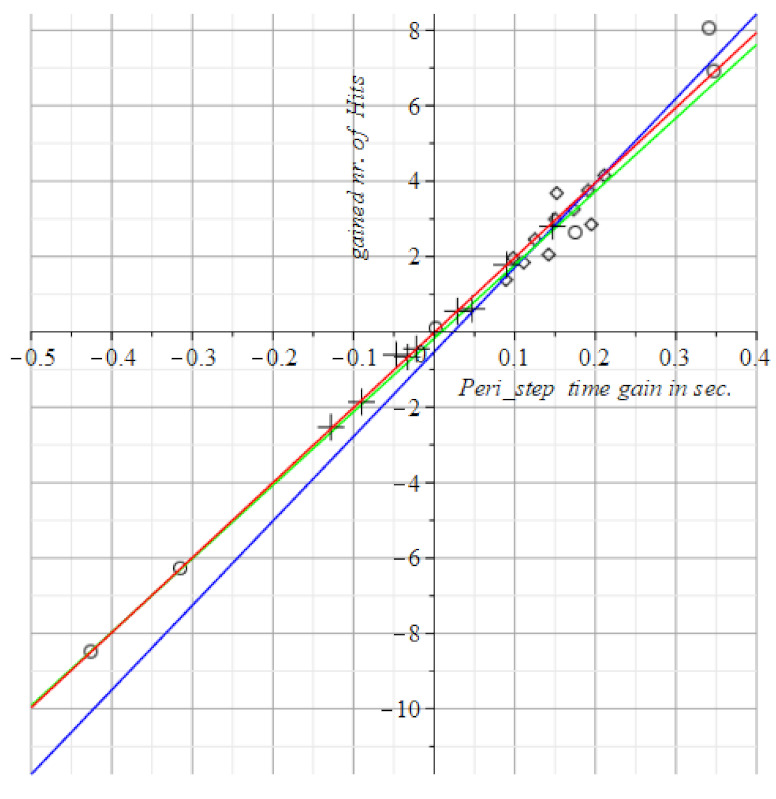
The results of the peripheral step test. The results discriminate between the motor group and the visuo-spatial-motor group: Three subjects in the motor group improved performance, one remained neutral (=the circle at the origin) and two scored less than prior to the motor intervention. The control group also shows a mixed picture. In the visuo-spatial-motor group all subjects improved performance. Circles and green line: Motor group, $T_{1}=$ −1.73 (±0.30) + 19.5 (±0.99) $T_{0}$. Rhombus pts., blue: Visuo-spatial-motor group, $T_{1}=$ −0.52 (±0.67) + 22.4 (±2.4) $T_{0}$. Plus signs, red line: Control group, $T_{1}=$ −0.02 (±0.06) + 19.9 (±0.67) $T_{0}$.

**Figure 11 geriatrics-06-00066-f011:**
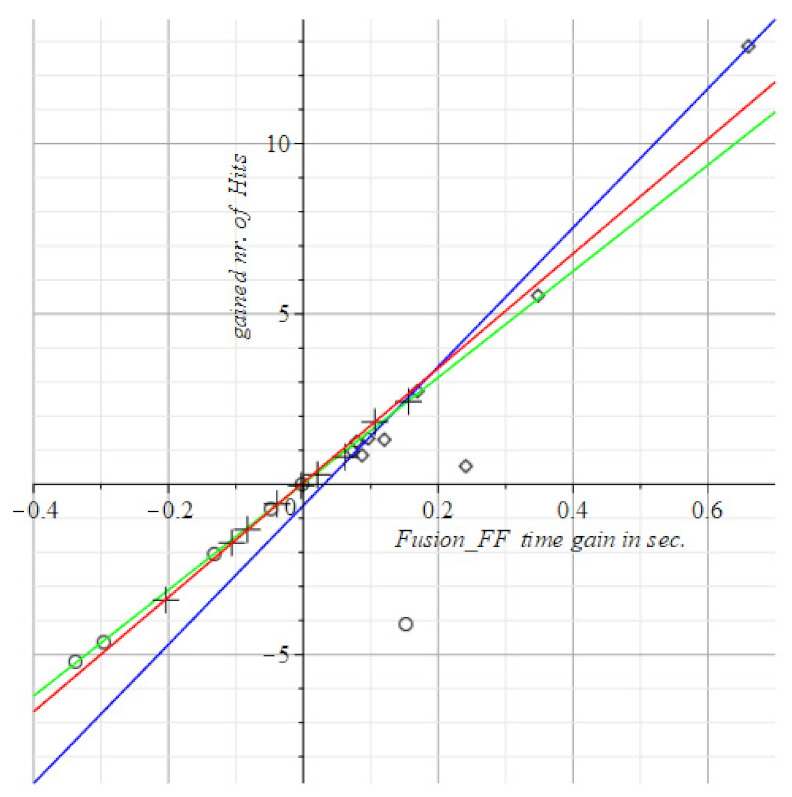
This depicts the results of the Fusion Flexibility test. The control group shows subjects with and without improvement; the motor group’s subjects performance declined while in the visuo-spatial-motor group all subjects show improvement. Circles: Motor intervention, green $T_{1}=$ 0.02 (±1.04) + 15.6 (±5.15) $T_{0}$. Rhombus pts: Visuo-spatial-motor intervention, blue $T_{1}=$ −0.63 (±0.67) + 20.4 (±2.4) $T_{0}$. Plus signs: The control group, red $T_{1}=$ 0.05 (±0.03) + 16.8 (±0.31) $T_{0}$.

**Table 1 geriatrics-06-00066-t001:** The mean pre-test measurements $T_{0}$ for the three groups ^1^.

SPPB	TUG	POMA
7.67 (±2.65)	8.66 (±2.60)	24.2 (±3.07)
5.17 (±2.64)	10.3 (±5.51)	20.7 (±4.50)
8.44 (±2.01)	7.40 (±2.23)	24.1 (±2.93)

^1^ The rows are: Visuo-spatial-motor group, pure motor group, and control group.

**Table 2 geriatrics-06-00066-t002:** The mean post-test measurements $T_{1}$ for the three groups ^1^.

SPPB	TUG	POMA
10.8 (±1.30)	6.90 (±1.95)	26.2 (±2.82)
9.00 (±2.53)	9.62 (±3.89)	24.2 (±3.19)
10.8 (±1.72)	6.16 (±1.81)	26.0 (±1.32)

^1^ The rows are: Visuo-spatial-motor group, pure motor group, and control group.

**Table 3 geriatrics-06-00066-t003:** Overview of outcomes of group independence testing for the Visuo-spatial-motor tests.

Test	Statistic	Critical Value	Probability
FitLight^®^Static	0.542256	12.5916	0.997286
FitLight^®^Dyn.	0.838832	12.5916	0.990992
Periph. Step	0.148314	12.5916	0.999936
Fusion Flex.	0.345730	12.5916	0.999243

All four groups have the same outcome: This statistical test does not provide enough evidence to conclude that the null hypothesis of similar groups is false. Or conclusion in other words; for the visuo-spatial-motor tests, is random allocation of subjects to the groups not refuted. Therefore: the groups are independent.

## Data Availability

All data are available in Excel Document sheets (the numerical data) and in Maple (the statistics) from: Koppelaar.Henk@GMail.com.

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
