# Peer review of "Proof of Concept of Novel Visuo-Spatial-Motor Fall Prevention Training for Old People"

_geriatrics, 2021, doi:10.3390/geriatrics6030066_

Round 1

Reviewer 1 Report

This paper does not follow the rules for randomized controlled trials (consort).

From the introduction the distinction between diagnosis and program effectiveness is not clear.

For the motor tests, even the standard deviations are not given with the means.

The different parts of the article are not respected. In materials and methods we find paragraph about literature ,discussion or results.

Despite the interest of this topic, many others examples could be given of the insufficiency of this article.

This paper cannot be published in this form.

Author Response

Esteemed Referee, thank you, for your help by your critics and comments. Below we repeat shortcuts of your remarks and answer what we did about it, per remark:

This paper does not follow the rules for randomized control:

The kind of study is not a clinical trial but an interventional study as now said at the beginning of section 2.7. To sustain its appropriateness, the test of statistical independence of the three intervention groups is in section 2.7.1. according to the multinomial (parallel) method advocated by Hogg and  Craig, Introduction to Mathematical Statistics; 7th ed.

From the introduction is the difference between diagnosis and program effectiveness not clear:

The introduction has been curated by one of us. The remaining text, organized via topics, clarifies the work.

For the motor test, the std deviations are not given:

Improvement: the standard deviations are inserted in the two tables of the Motor tests.

The different parts of the article are not respected

This fault is repaired as follows, in Materials and Methods:

  1. previous lines 251 – 260 are severely reduced and partially put in 1.1 in Best test Selection;
  2. previous lines 290 – 297 are severely reduced by deleting most of the summary of Nemoto’s research;
  3. Sections 2.5.1, 2.5.2 in Research Development is severely reduced and if appropriate moved to the Discussion or Literature;
  4. Over thirty additional improvements are made;

Despite the interest of the topic, many other examples of deficiencies could be given:

This fault is repaired as follows:

  1. Additional iIntroduction of improvements as requested by Reviewer 2: see below;
  2. Typo’s were rare, but have been reaired;
  3. Text parts (and including references) have been curated by deletion;
  4. Explanations are added if needed to clarify an argument.

This is not all, because Reviewer 2 has also many not all overlapping criticism,s.  This gave us opportunity to additional improvements to the submission.

Henk Koppelaar

Reviewer 2 Report

A multidisciplinary team 1) reviews the current strategies to prevent falls in the older adults and 2) provides extensive proof-of-concept of a visuo-spatial-motor training successful used to enhance athlete's abilities is translated to be used in old people to prevent falling.

The abstract should be rewritten to provide a better explanation of the current problem, that the present work provides a review on the current intervention and finally provides proof of concept on the effectiveness of VSM training. Sentences are not engaging. The abstract should be densified in information to provide a trustful argument that reflects the ms' extensive contents. As it stands right not does not play the role of a summary.

Title. The title should reflect the goal: providing proof-of-concept of a VSM training already used in sports translated to gerontology/geriatrics. Now, as it stands, one does not know what's the paper about.

  1. We explain, We take, we do, etc Please, use the scientific form ' x is explained', etc when possible

  2. Please, provide references to support your statement.

  1. 'this paper many papers'. Please, improve the sentence

  1. and other parts. Please, note that 'elderly' and 'elderly people' are considered ageists terms. Therefore, 'old people' 'older adults' or 'older people' should be used, instead.

74.82. A whole paragraph to discuss billionaire costs is no needed, mostly when it concerns to an age where the quality of life is one of the most important indicators. Please, short this part and add some data on psychological impact and well-being to complement.

A graphical abstract would benefit the conceptual description of the introduction and help to locate the proposal.

605 The authors state that 'We did not group or adjust 605 for age and sex or corresponding baseline values of data.' Still, they could briefly discuss if there were items of analysis that would show or not an age or sex effect that will encourage future studies/analysis.

Since the figures plot the data, the symbols could provide information of who is male/female, so at least this is informative

Author Response

Esteemed Referee, thank you for your help by your critics and extensive comments. Below we repeat shortcuts of your remarks and answer what we did about it, per remark:

Your remarks about Multidisciplinary team and the Abstract: one of us Dr. P.K-M rewrote the Abstract to satisfy your quite rightly remarks, also to us authors’ full satisfaction.

Title:  The title is extended with help of your remark.

26   :  To replace ‘we do’, ‘we explain’:  The passive form is now used in 32 more places than in previous version.

27   :   By rewriting most of this part, the appropriate referencing should be included.

33   :   The error ‘This paper many papers … ‘  is removed.

56   :   According to your suggestion: our obsolete term ‘elderly’ is replaced everywhere by ‘older adults’.

74.82 :  The paragraph about the insurers billionaire cost is shortened and the impact to psychological well-being is inserted, by many papers referring to the mental impact of fear of falling.

A Graphical Abstract is added.

605   :   An observation about the subjects has been added in lines 878 – 879.

Grouping for sex. Male subjects were volunteering for the novel visuo-motor training.
The color red is given to male data points of the three Motor Tests (SPPS, POMA and TUG) in Figures 5 – 7. All other subjects are female. The red data points are not yet introduced in Figures 8 – 11 of the four Visuo-spatial test, due to a software error. This will be repaired in the final submission.

This is not all, because Reviewer 1 has also many not all overlapping criticisms.  This gave us an opportunity to additional improvements to the submission.

Round 2

Reviewer 1 Report

This new version can be published

Author Response

Thank you for your work. It greatly helped to improve the paper.

Henk Koppelaar

Reviewer 2 Report

The authors have modified the Ms and have provided answers to the questions. In my opinion, the revised version of the Ms now fits better with the study done, is more informative.

Please, check line 22 an elderly is still there.

Author Response

(The authors gave the same response as above.)
